# A Bifunctional Pt/CeO₂-Cu₁/CeO₂ Catalyst System for Isooctane Oxidation under Fully Simulated Engine-Exhaust Condition: Eliminating the Inhibition by CO

Fan Lin [1], Carlos E. García-Vargas [2] and Yong Wang [1,2,*]

[1] Institute for Integrated Catalysis, Pacific Northwest National Laboratory, P.O. Box 999, Richland, WA 99354, USA

[2] Voiland School of Chemical Engineering and Bioengineering, Washington State University, Pullman, WA 99163, USA

* Correspondence: yong.wang@pnnl.gov

**Abstract:** Pt-based catalysts, because of their outstanding activity for hydrocarbon oxidation, are widely used in the engine-exhaust aftertreatment system to remove hydrocarbon emissions. However, the CO and $NO_x$ present in real engine exhausts compete with hydrocarbons for active Pt sites, and thus inhibit hydrocarbon oxidation. In this work, we evaluated the inhibition effects of CO and NO on isooctane oxidation over a Pt/CeO₂ catalyst under the simulated condition of the US DRIVE test protocol (S-GDI, stoichiometric gasoline direct injection). We also leveraged a low-cost single-atom Cu₁/CeO₂ catalyst, which is highly active for low-temperature CO oxidation, to eliminate the inhibition effect of CO. Specifically, by physically mixing Cu₁/CeO₂ and Pt/CeO₂, all the CO is completely converted below 200 °C under simulated exhaust condition, which helps lower the isooctane oxidation temperature. However, the unconverted NO still strongly suppresses HC oxidation. Possible strategies to address the NO inhibitor were proposed.

**Keywords:** emission control; hydrocarbon oxidation; Pt/CeO₂; Cu single-atom catalyst

## 1. Introduction

Unburned hydrocarbon fuels are one of the major pollutants in the exhaust of internal combustion engines [1,2]. Although hydrocarbon emissions can be addressed by the three-way catalysts (TWCs) or the diesel oxidation catalysts (DOCs), the low-temperature hydrocarbon oxidation during the "cold start" period is still a challenging task for the engine aftertreatment system [3], due to stable C-H bonds, especially for the saturated alkane molecules [4]. Thus, low-temperature hydrocarbon oxidation catalysts are needed.

So far, the expensive platinum group metals (PGM), such as Pt and Pd, are still the most efficient catalysts for low-temperature hydrocarbon oxidation, due to their ability to activate C-H bonds of hydrocarbon molecules [5]. However, under real exhaust conditions, the catalytic reaction of hydrocarbons on PGMs can be inhibited by other components, including CO, $NO_x$, and $H_2O$, due to the competitive adsorption [6–12]. For instance, $CH_4$ oxidation on Pd catalysts is strongly inhibited by $H_2O$, associated with the formation of OH species in the active PdO phase [10–12]. In contrast, for the Pt-based catalyst, the inhibition effect of $H_2O$ is weaker, with CO and NO, and HC itself as the major inhibitor. A typical empirical kinetic model for hydrocarbon (HC) oxidation on Pt-based catalysts is given in the form of [6]:

$$r_{HC} = \frac{k Y_{HC} Y_{O_2}}{T^n (1 + K_{CO} Y_{CO} + K_{HC} Y_{HC} + K_{NO} Y_{NO})^2} \tag{1}$$

where $r_{HC}$ denotes the rate of hydrocarbon oxidation; $T$ is the temperature; $k$ is the rate constant; $Y_i$ represents the concentration of component $i$ ($i$ = HC, $O_2$, CO, or NO); and $K_i$ represents the inhibition factor of the component $i$ ($i$ = HC, CO, or NO). The inhibition

factor, $K_i$, is essentially associated with the strength of molecule adsorption on the metal catalyst. Among the hydrocarbon molecules, the saturated alkanes were adsorbed more weakly on metal catalysts than the unsaturated alkenes or aromatics, and thus the oxidation of alkanes is more easily inhibited by other components [9].

In a previous work [13], we synthesized a Pt/CeO$_2$ catalyst (Table 1), which showed improved hydrocarbon-oxidation activity, by using a CeO$_2$ support pre-calcined at 800 °C. Loading Pt on this high-temperature pretreated CeO$_2$ formed two-dimensional Pt rafts, which can be turned into three-dimensional Pt nanoparticles (1–1.5 nm) upon reduction treatment (by CO at 250 °C). On one hand, the Pt rafts or nanoparticles are responsible for activating HC molecules. On the other hand, the high-temperature pretreatment decreased the surface defects on the CeO$_2$ and thus enhanced the mobility of the surface lattice oxygen and promoted the redox activity of the catalyst, regardless of the Pt cluster morphology. This improved redox activity promoted the catalytic activity of Pt/CeO$_2$ for HC oxidation, for both weakly adsorbed alkanes and strongly adsorbed aromatics and alkenes, under both lean and stoichiometric conditions. However, the fully simulated exhaust can dramatically decrease the reactivity of hydrocarbon molecules, due to the competitive adsorption of CO and NO$_x$, especially for the weakly adsorbed alkanes. For instance, $T_{50}$ (temperature for 50% conversion) for isooctane oxidation on this Pt/CeO$_2$ catalyst was as low as 161 and 194 °C under rich (0.74% O$_2$) and lean (10% O$_2$) conditions (WHSV = 200L h$^{-1}$ g$^{-1}$), respectively. However, under fully simulated conditions (S-GDI and LTC-G), the presence of CO, NO$_x$, and H$_2$O significantly inhibited isooctane oxidation, increasing the $T_{50}$ to 257 and 238 °C, respectively. Therefore, in addition to improving the intrinsic activity of the oxidation catalysts, eliminating the inhibition effects of other exhaust components is also a key to achieving low-temperature hydrocarbon oxidation reactivity under real exhaust conditions.

**Table 1.** Properties of Pt/CeO$_2$ and Cu$_1$/CeO$_2$ catalysts.

| | **Pt/CeO$_2$ [13]** | **Cu$_1$/CeO$_2$ [14]** |
|---|---|---|
| Metal Loading | 1.66 wt.% Pt [a] | 2 wt.% Cu [b] |
| Surface Area (m$^2$ g$^{-1}$) | 30 | 46 |

[a] measured by ICP, [b] nominal loading.

Introducing catalytic sites which are specifically active for CO or NO conversion at lower temperatures (lower than the onset temperature for hydrocarbon oxidation) is a strategy to eliminate their inhibition effects for hydrocarbon oxidation. In our previous work [14], a low cost Cu$_1$/CeO$_2$ single-atom catalyst was developed for low-temperature CO oxidation. The Cu$_1$/CeO$_2$ with 2 wt.% Cu loading (Table 1) was synthesized via the atom trapping method (800 °C calcination) [15]. The Cu single-atom active sites trapped at step defects, with the Cu−O−Ce being the primary active oxygen species, can activate either lattice or adatom oxygen atoms for CO oxidation, accessing additional reaction channels as the catalyst environment changes. This Cu$_1$/CeO$_2$ catalyst is specifically active for CO oxidation, with low activity for HC oxidation. The complete CO conversion was achieved below 150 °C on this Cu$_1$/CeO$_2$ catalyst under the condition of 1% CO/8% O$_2$ with a GHSV of 300,000 h$^{-1}$ [14]. Even under the fully simulated condition of the CDC (clean diesel combustion)-test protocol (12%O$_2$, 6% H$_2$O, 6% CO$_2$, 500 ppm CO, 100 ppm H$_2$, 200 ppm NO, 500 ppm C$_3$H$_6$, 300 ppm C$_3$H$_6$, 100 ppm C$_3$H$_8$) [16], this catalyst showed a low $T_{90}$ of 180 °C for CO conversion. In addition, strong anchoring of Cu to step defects effectively halts sintering and deactivation.

In this work, we systematically analyzed the inhibition effects of CO, NO, and H$_2$O on the isooctane oxidation over Pt/CeO$_2$ catalysts under the stoichiometric condition. We also leveraged the low-cost Cu$_1$/CeO$_2$ catalyst with high CO oxidation activity to eliminate the CO inhibition effect. It was found that CO and NO are the major inhibitors for isooctane oxidation under fully simulated exhaust conditions. Although the CO inhibitor was successfully addressed by Cu$_1$/CeO$_2$, the isooctane oxidation on Pt/CeO$_2$ was still strongly

inhibited by NO in the exhaust. We also proposed possible strategies for eliminating the inhibition effect of NO to further improve hydrocarbon oxidation reactivity under real exhaust conditions.

## 2. Results

### 2.1. Inhibition of HC Oxidation over Pt/CeO$_2$ Catalysts by CO and NO

We first evaluated the respective inhibition effect of CO, NO, and H$_2$O on the oxidation of isooctane (C$_8$H$_{18}$) under stoichiometric conditions. The test conditions are summarized in Table 2. The simulated exhaust conditions are based on the U.S. DRIVE protocol S-GDI (stoichiometric gasoline direct injection) [16]. The concentrations of CO, NO, and H$_2$ were adjusted from the original S-GDI protocol, due to the limitation of the available gas-mixture source (5000 ppm CO/2000 ppm NO/1000 ppm H$_2$).

**Table 2.** Gas composition of the fully simulated S-GDI and simplified conditions for catalyst tests [#].

| Gas Composition | O$_2$ | O$_2$ + H$_2$O | O$_2$ + NO | O$_2$ + CO | S-GDI |
|---|---|---|---|---|---|
| O$_2$ | 0.74% | 0.74% | 0.74% | 0.74% | 0.74% |
| H$_2$O | 0 | 13% | 0 | 0 | 13% |
| CO$_2$ | 0 | 0 | 0 | 0 | 13% |
| CO | 0 | 0 | 0 | 3500 ppm | 3500 ppm |
| NO | 0 | 0 | 1400 ppm | 0 | 1400 ppm |
| H$_2$ | 0 | 0 | 0 | 0 | 700 ppm |
| Isooctane | 3000 C$_1$ | 3000 C$_1$ | 3000 C$_1$ | 3000 C$_1$ | 3000 C$_1$ |

[#] Space velocity: 200 L h$^{-1}$ g$_{Pt/CeO2}$$^{-1}$.

Figure 1 shows the light-off curves of isooctane oxidation on the Pt/CeO$_2$ catalyst under stoichiometric conditions. As shown in Figure 1a, under the simplified O$_2$-only condition (0.74% O$_2$), the oxidation of isooctane on Pt/CeO$_2$ was initiated at around 150 °C and the complete conversion was achieved below 200 °C. However, under the fully simulated S-GDI condition, the isooctane light-off temperature was significantly increased, by almost 100 °C. For instance, $T_{50}$ of isooctane increases from 161 °C under the O$_2$-only condition to 257 °C under the S-GDI condition, as shown in Figure 1d. It seems that the decrease in isooctane oxidation reactivity is due to the inhibition effects of other components in the fully simulated exhaust condition.

CO, NO, and H$_2$O are all known to be potential inhibitors for hydrocarbon oxidation on noble metal catalysts. Therefore, we evaluated the inhibition effect of these three components by testing the light-off of isooctane oxidation in the O$_2$ + H$_2$O, O$_2$ + CO, and O$_2$ + NO conditions (composition listed in Table 2), separately. As shown in Figure 1a, H$_2$O only slightly increased the temperature of isooctane oxidation ($T_{50}$ = 177 °C, Figure 1d), whereas CO and NO both significantly retarded the light-off of isooctane ($T_{50}$ = 237–238 °C), close to the case of the S-GDI condition. These results suggest that CO and NO play a major role in inhibiting isooctane oxidation on the Pt/CeO$_2$ catalysts.

In contrast to isooctane, the oxidation reactivities of NO and CO on Pt/CeO$_2$ were not significantly affected by the presence of other components (e.g., CO, NO, and H$_2$O), as shown by the light-off curves of NO (Figure 1b) and CO (Figure 1c) under different conditions. As shown in Figure 1c, CO oxidation was initiated at a temperature as low as 150 °C, and reached 50% conversion at 217–231 °C. These $T_{50}$ values of CO conversion are slightly lower than the temperature for the initiation of isooctane light-off under O$_2$ + CO and S-GID conditions (225–245 °C), as shown in Figure 1a. The sequential conversions of CO and isooctane suggest that the onset of isooctane oxidation requires the removal of the inhibitor CO.

On the other hand, Pt/CeO$_2$ has a relatively low activity for NO conversion. As shown in Figure 1b, the onset for NO conversion was at around 225 °C, and reached a maximum of about 30–40% at around 275 °C. While NO was converted into NO$_2$ and N$_2$O, a portion of NO$_x$ (NO, NO$_2$ and N$_2$O) also underwent deNOx reactions (forming N$_2$), as indicated by

the total-$NO_x$ (NO, $NO_2$ and $N_2O$) conversion curve. Therefore, the NO inhibitor cannot be completely removed throughout the isooctane light-off test, due to the low NO conversion.

Given that CO and NO are the major inhibitors for isooctane oxidation on $Pt/CeO_2$ in the fully simulated exhaust, we proposed to introduce additional catalyst components which can remove these inhibitors at a lower temperature, so that the isooctane oxidation temperature on $Pt/CeO_2$ can be significantly reduced, ideally to the level of the $O_2 + H_2O$ condition.

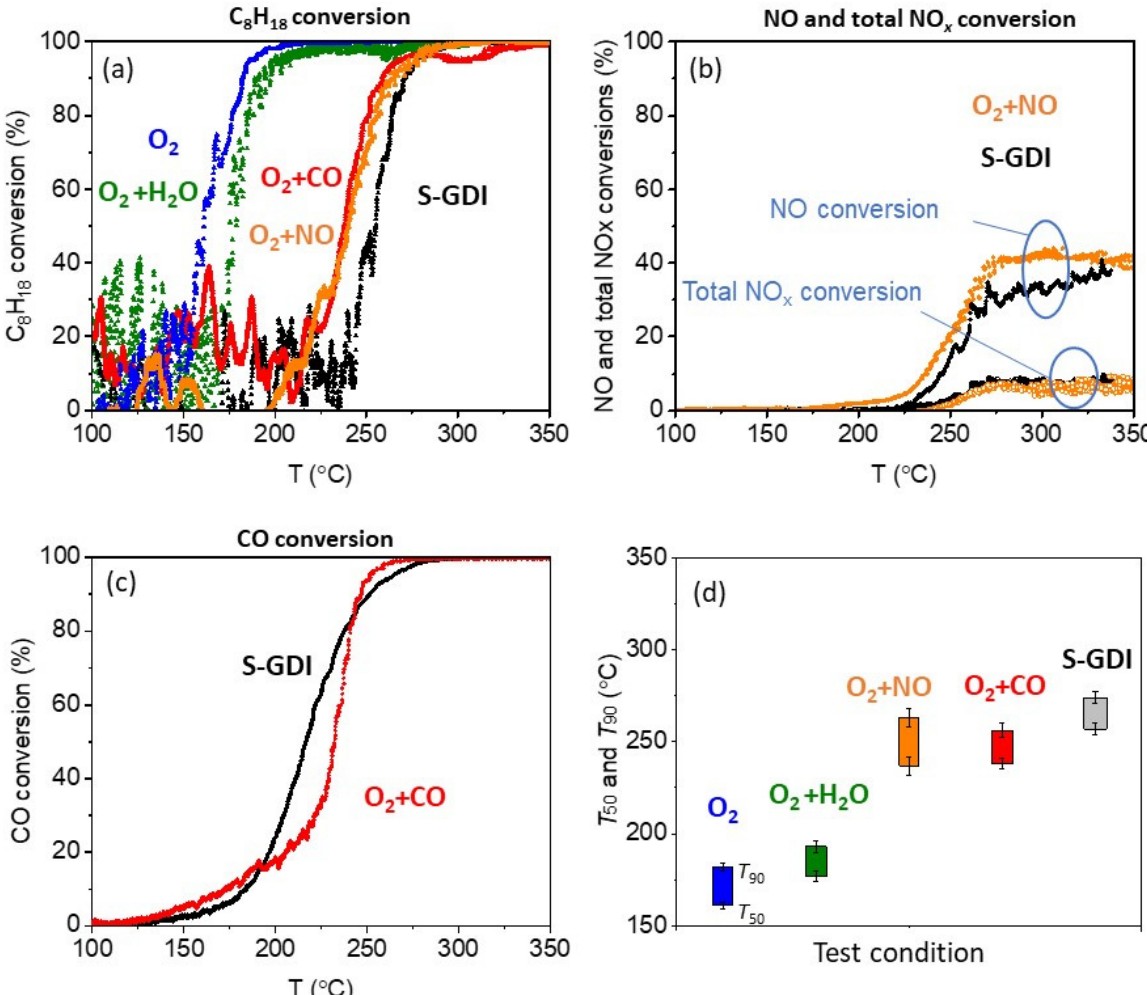

**Figure 1.** Light-off curves of isooctane (**a**), NO (**b**), and CO (**c**) during isooctane oxidation on $Pt/CeO_2$ catalysts under fully simulated stoichiometric condition (S-GDI) and various simplified exhaust conditions (space velocity: 200L $h^{-1}$ $g_{Pt/CeO2}^{-1}$); (**d**) summary of $T_{50}$ and $T_{90}$ for isooctane conversion.

### 2.2. Eliminating the CO-Inhibition Effect Using $Cu_1/CeO_2$ Single-Atom Catalyst

In recent work by our group [14], a $CeO_2$-supported Cu single-atom catalyst $Cu_1/CeO_2$, as a low-cost non-PGM catalyst, was found to be a highly active and stable catalyst for low-temperature CO oxidation. Therefore, we introduced the $Cu_1/CeO_2$ to eliminate the inhibitor CO at low temperature by physically mixing $Cu_1/CeO_2$ with $Pt/CeO_2$. Figure 2 shows the performance of the $Pt/CeO_2$-$Cu_1/CeO_2$ bifunctional catalyst system, evaluated under a Pt-based space velocity of 200L $h^{-1}$ $g_{Pt/CeO2}^{-1}$, similar to the $Pt/CeO_2$ catalyst test of Figure 1.

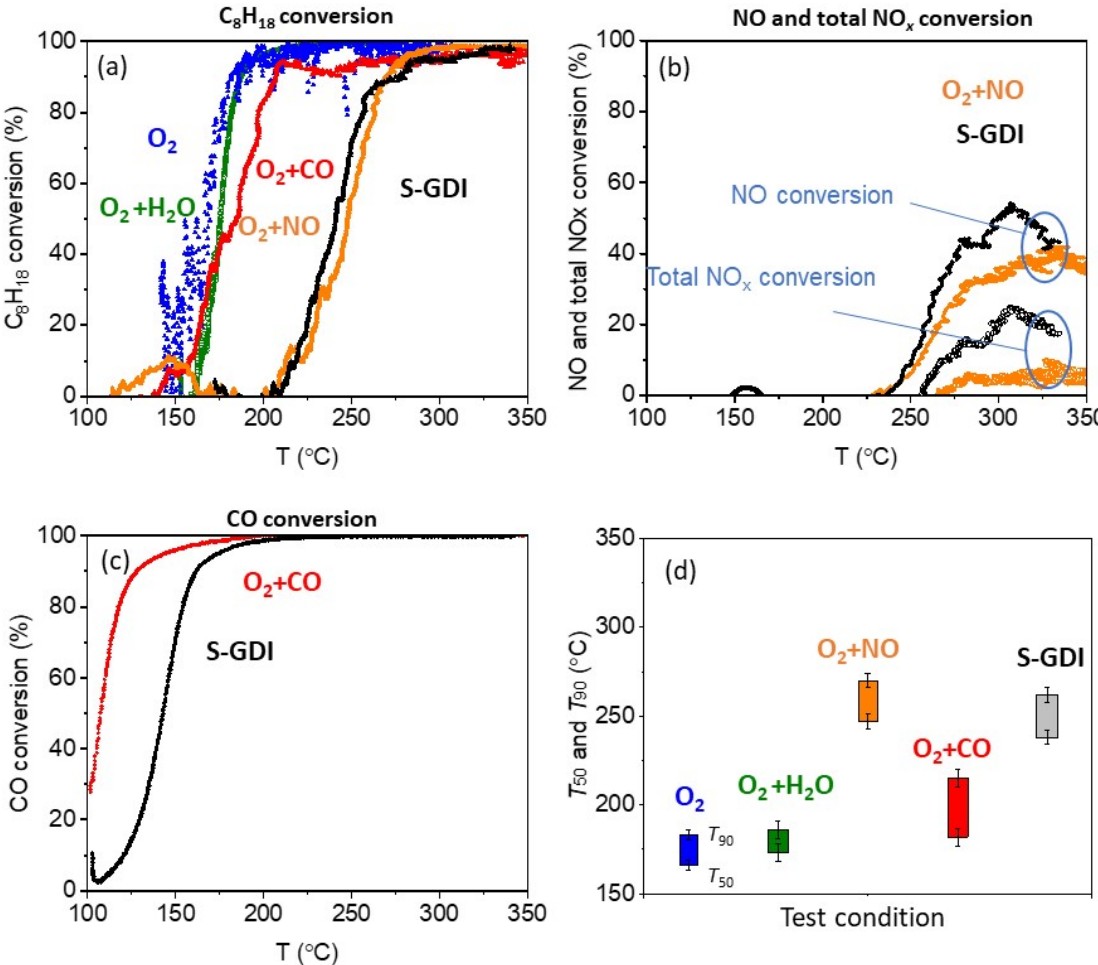

**Figure 2.** Light-off curves of isooctane (**a**), NO (**b**), and CO (**c**) on $Pt/CeO_2$ + $Cu_1/CeO_2$ catalyst mixture (1:1 weight ratio) under fully simulated stoichiometric condition (S-GDI) and various simplified exhaust conditions (space velocity: 200L $h^{-1}$ $g_{Pt/CeO2}^{-1}$); (**d**) summary of $T_{50}$ and $T_{90}$ for isooctane conversion.

The isooctane light-off curves are shown in Figure 2a. In the $O_2$-only condition, the isooctane oxidation reactivity on the $Cu_1/CeO_2$-$Pt/CeO_2$ mixture is essentially the same as on $Pt/CeO_2$ (Figure 1a), because the $Cu_1/CeO_2$ single-atom catalyst has very low activity for hydrocarbon oxidation. For instance, no isooctane oxidation was observed on $Cu_1/CeO_2$ below 300 °C in the S-GDI condition. The Introduction of 13% of $H_2O$ ($O_2$ + $H_2O$ condition) has very weak inhibition effect on isooctane oxidation (Figure 2a), which is also similar to the $Pt/CeO_2$ catalyst. In the $O_2$ + CO condition, the $Cu_1/CeO_2$-$Pt/CeO_2$ mixture shows improved isooctane-oxidation activity in comparison to $Pt/CeO_2$, with $T_{50}$ decreasing significantly, from 238 °C to 182 °C, because $Cu_1/CeO_2$ completely converts CO below 170 °C (Figure 2c) to eliminate the inhibition effect of CO. However, in the $O_2$ + NO condition, the $Cu_1/CeO_2$-$Pt/CeO_2$ mixture does not improve the isooctane oxidation activity, as $Cu_1/CeO_2$ is not active for NO conversion (Figure 2b). Consequently, in the fully simulated S-GDI condition, the $Cu_1/CeO_2$-$Pt/CeO_2$ mixture has only limited improvement for isooctane oxidation in comparison to $Pt/CeO_2$, with $T_{50}$ decreasing slightly, from 257 °C to 238 °C. This small improvement is likely ascribed to the elimination of the CO inhibitor below 200 °C.

It is noted that the CO oxidation activity of the $Cu_1/CeO_2$-$Pt/CeO_2$ mixture is lower under S-GDI than the $O_2$ + CO conditions (Figure 2c). This is likely because NO or $H_2O$ has some inhibition effect on CO oxidation over $Cu_1/CeO_2$ [14]. The comparison of performance under the $O_2$ + NO + CO (0.75% $O_2$ + 1400 ppm NO + 3500 ppm CO) and

S-GDI conditions (Figure S1) indicates that $H_2O$ decreases both the CO oxidation activity and NO conversion activity of the $Pt/CeO_2 + Cu_1/CeO_2$ mixture, but barely affects its isooctane oxidation activity. Nevertheless, $Cu_1/CeO_2$ is capable of completely converting CO below 200 °C under the S-GDI condition.

On the other hand, for the PGM catalysts, not only does CO inhibit the oxidation of HCs, but HCs can also inhibit the oxidation of CO [6–9]. In contrast, on the $Cu_1/CeO_2$ catalyst, we noted that the presence of isooctane did not significantly change the CO oxidation activity. $T_{50}$ and $T_{90}$ for CO oxidation on $Cu_1/CeO_2$-$Pt/CeO_2$ in the mixture of isooctane (3000 ppm $C_1$) + CO (3500 ppm) + $O_2$ (0.74%) are 105 and 128 °C, respectively (Figure 2c), close to those reported for $Cu_1/CeO_2$ in the feed of CO (1%) + $O_2$ (4%) ($T_{50} = 96$ and $T_{90} = 135$ °C, Figure 3). This suggests that the $Cu_1/CeO_2$ single-atom catalyst also overcomes the inhibition effect of HCs on the CO oxidation reaction, because the H-C group of the HC molecules adsorbed more weakly on the Cu-O-Ce sites in comparison to the PMG sites [17]. This feature makes $Cu_1/CeO_2$ an attractive catalyst and specifically active for CO oxidation.

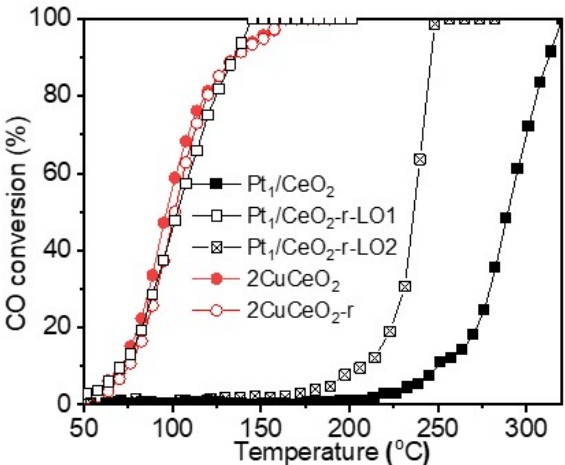

**Figure 3.** Light-off curves of CO oxidation on the pristine and reduced $Cu_1/CeO_2$ and $Pt_1/CeO_2$ catalysts ("-r" denotes reduced catalysts; "-LO1" and "-LO2" denote 1st and 2nd light-off test, respectively; light-off test condition: $1\%CO/4\%O_2/Ar$ balance, 20 mg of catalysts, 500 mg of inert SiC and 3 °C/min at 300 L $(g_{cat} h)^{-1}$; reduction-treatment condition: 1% CO/Ar, 275 °C).

## 2.3. Stability of $Cu_1/CeO_2$ Single-Atom Catalyst

Another advantage of the $Cu_1/CeO_2$ catalyst is its outstanding stability in comparison with the Pt-based catalysts. It is known that the noble metal single-atom catalysts (e.g., $Pt_1/CeO_2$ and $Pd_1/CeO_2$) easily undergo metal aggregation under reduction conditions [18]. In contrast, Cu single-atom sites on $CeO_2$ are highly resistant to reduction conditions with CO [14]. Figure 3 compares the CO oxidation activities of the pristine and reduced $Cu_1/CeO_2$ and $Pt_1/CeO_2$ catalysts, both of which were synthesized by the atomic-trapping method. The pristine $Pt_1/CeO_2$ has relatively low activity for CO oxidation under 1% CO and 4% $O_2$ conditions, with a high $T_{50}$ of 290 °C. Upon reduction treatment by 1% CO at 275 °C, Pt single-atom sites on the $Pt_1/CeO_2$-reduced sample aggregated to form metallic Pt nanoparticles ($Pt_{NP}$). These metallic $Pt_{NP}$ sites are in fact very active for CO oxidation, and show a $T_{50}$ as low as 102 °C. However, the metallic Pt sites on $Pt_{NP}$ are not stable. After one cycle of light-off tests to >300 °C, due to the surface oxidation or Pt-atom redispersion, the activity of the $Pt_1/CeO_2$-reduced sample was degraded in the second light-off. In contrast, the pristine and reduced $Cu_1/CeO_2$ present the same high activity for CO oxidation ($T_{50} = 100$–102 °C), indicating that the reduction atmosphere did not cause any change on the Cu single-atom sites.

## 3. Materials and Methods

The $CeO_2$-supported Pt catalysts ($Pt/CeO_2$), with improved hydrocarbon oxidation activity, were prepared by the incipient-wetness-impregnation method, as described in our previous work [13]. A commercial $CeO_2$ (99.99%, 10 nm, US Research Nanomaterials, Inc., Houston, TX, USA) pre-calcined at 800 °C for 10 h was used as the support. Aqueous solution of $[Pt(NH_3)_4](NO_3)_2$ (99.995%, Sigma Aldrich, St. Louis, MO, USA) was impregnated on the $CeO_2$ support to achieve a nominal 2 wt.% Pt loading. The Pt-loaded $CeO_2$ was dried at 70 °C for 4 h and then calcined at 500 °C for 4 h in flowing air to obtain the $Pt/CeO_2$ catalysts. The actual Pt content was determined by ICP (inductively coupled plasma) analysis. The detailed characterization results of the $Pt/CeO_2$ catalyst are available in [13].

The $CeO_2$-supported Cu single-atom catalyst ($Cu_1/CeO_2$) with 2wt.% Cu loading was prepared by the incipient-wetness-impregnation method and high-temperature calcination, as described in [14]. The $CeO_2$ support was prepared by thermal decomposition of $Ce(NO_3)_3 \cdot 6H_2O$ (99%, from Sigma-Aldrich) in air at 350 °C for 2 h. After loading $Cu(NO_3)_2 \cdot 2.5H_2O$ (Sigma-Aldrich) on the $CeO_2$ support by incipient wetness impregnation, the catalyst was dried in air at 150 °C for 1 h, followed by 800 °C calcination for 10 h. The detailed characterization results of this $Cu_1/CeO_2$ catalyst are available in [14]. A $Pt_1/CeO_2$ single-atom catalyst (with 1 wt.% Pt loading), as a control catalyst for $Cu_1/CeO_2$, was synthesized with the same method, using $[Pt(NH_3)_4](NO_3)_2$ as the Pt precursor. The $Pt/CeO_2$-$Cu_1/CeO_2$ catalyst mixture (1:1 mass ratio) was prepared by grinding the two catalysts together using a mortar and pestle.

The catalyst performance for isooctane oxidation was evaluated by temperature-programed reaction using a plug flow micro-reactor system, with the reactor effluent composition quantified using an FTIR gas analyzer (MKS, Multigas 2030). The catalyst powders were pressed, crushed, and sieved (180–250 μm) for the tests. A total 150 mg of the $Pt/CeO_2$ catalyst (or 300 mg of the $Pt/CeO_2$-$Cu/CeO_2$ catalyst mixture) was mixed with 400 mg of SiC diluent and loaded in a quartz reactor (inner diameter ~10 mm) above a porous frit. The catalysts were tested with different gas compositions to evaluate the inhibition effects of different exhaust components (CO, NO, $H_2O$). The specific test conditions are listed in Table 2. The fully simulated exhaust conditions are based on the U.S. DRIVE protocols S-GDI (stoichiometric gasoline direct injection) [16]. The concentrations of CO, NO, and $H_2$ were adjusted from the original protocols based on the available gas-mixture source (5000 ppm CO/2000 ppm NO/1000 ppm $H_2$). The liquid isooctane and $H_2O$ were introduced into vaporization zones located at the upstream of the reactor using a syringe infusion pump and an HPLC pump, respectively.

To evaluate and compare the stabilities of the $Cu_1/CeO_2$ and $Pt_1/CeO_2$ single-atom catalysts under the reduction condition, the pristine catalyst samples were pretreated in 1%CO/Ar at 275 °C for 30min. The CO-reduced catalysts were denoted as $Cu_1/CeO_2$-r and $Pt_1/CeO_2$-r, respectively. The CO-oxidation activities of the pristine and reduced catalysts were test by CO light-off experiments with a simplified condition (1%CO/4%$O_2$/Ar balance, 3 °C min$^{-1}$, space velocity 300 L ($g_{cat}$ h)$^{-1}$), using a mixture of 20 mg catalysts and 500 mg SiC diluent. The catalysts were loaded into a 4.0-mm-i.d. × 40.64-cm-long quartz tube packed in-between inert quartz wool. The reactor effluent was analyzed using z GC2060 gas chromatograph model by Shanghai Ruimin GC Instruments, Inc.

## 4. Discussion

For the catalytic oxidation of isooctane on the $Pt/CeO_2$ catalyst under the fully simulated S-GDI condition, CO and NO were identified as the two major inhibitors in lowering the HC oxidation reactivity. These inhibition effects can be addressed by introducing catalytic sites or co-catalysts to preferentially convert CO or NO. The low-cost and stable $Cu_1/CeO_2$ single-atom catalyst, which is capable of completely converting CO below 200 °C under the fully simulated condition, can be added as a co-catalyst to eliminate the CO inhibitor. It is worth pointing out that the two types of catalytic sites, Cu single-atoms

sites and Pt clusters, were integrated via physical mixing rather than co-impregnation of two sites on the same $CeO_2$ support. This is to avoid interactions between these two types of sites, such as the formation of a Pt-Cu alloy or the coverage of Cu single-atom sites by Pt clusters. For instance, a bi-metallic $Pt/Cu_1/CeO_2$ catalyst, which was prepared by loading Pt clusters on an as-synthesized $Cu_1/CeO_2$ single-atom catalyst, showed lower isooctane oxidation activity than both the $Pt/CeO_2$ catalyst and the $Pt/CeO_2$-$Cu_1/CeO_2$ mixture, with equivalent Pt loading (Figure S2). However, this bi-functional $Pt/CeO_2$-$Cu_1/CeO_2$ catalyst system has low activity for NO conversion, and thus the isooctane oxidation reactivity was not obviously improved in the presence of NO.

Because $Pt/CeO_2$ is capable of converting isooctane below 200 °C under the "$O_2$ only" condition (0.75% $O_2$, Figure 1d), it would be ideal if NO could be eliminated below 200 °C so that the noble metal $Pt/CeO_2$ catalyst could be made best use of for HC oxidation. Potential methods for low-temperature NOx abatement include low-temperature HC-SCR (hydrocarbon-selective catalytic reduction), CO-NO reaction, and the use of PNA (passive NOx adsorber).

One potential approach is to incorporate low-temperature HC-SCR catalysts to utilize HCs to reduce $NO_x$. For instance, $Ag/Al_2O_3$ were reported to be active HC-SCR catalysts [19–21]. A wide range of HCs, including alkane, alkenes, and oxygenates, can serve as reductants for $NO_x$ reduction on $Ag/Al_2O_3$. Although the operation temperature windows (>80% NOx conversion) are typically between 350 °C and 500 °C [20], it was found that the presence of $H_2$ can significantly improve the activity of $Ag/Al_2O_3$ and decrease the $NO_x$ conversion temperature [21]. For example, on a 2 wt.%$Ag/Al_2O_3$ catalyst [19], >95% $NO_x$ conversion was achieved at 200 °C, with octane as the reductant, in exhaust steam with a high $H_2$/NO ratio (720 ppm NO, 570 ppm $C_8H_{18}$, 4.3% $O_2$; 7200 ppm $H_2$, 7.2% $H_2O$). The mechanisms for the promotion effect of $H_2$ on HC-SCR activity are still under debate. Generally, two types of mechanisms were proposed: (1) $H_2$ reducing the $Ag^+$ species into the more active $Ag^{\delta+}$ or $Ag^0$ sites [22] and/or (2) $H_2$ modifying the nature and quantity of reaction intermediates [21]. However, the real exhaust contains less $H_2$ (e.g., 1670 ppm $H_2$ and 1000 ppm NO for the S-GDI condition [16]), and thus an HC-SCR catalyst with high activity under low $H_2$ concentration is needed for this approach. Alternatively, $H_2$ sources can be added to the aftertreatment system to provide sufficient $H_2$ to facilitate low temperature HC-SCR. For instance, a gas fuel reformer can be used to generated $H_2$ on board [23].

Another possible strategy is to promote a low-temperature (CO + NO) reaction by introducing a $Rh_1/CeO_2$ single-atom catalyst recently reported by Khivantsev et al. [24]. In this work, a 0.5 wt.% $Rh_1/CeO_2$ catalyst was reported to complete the NO abatement above 120 °C in a (460 ppm NO + 1750 ppm CO) stream. This single Rh atom was demonstrated to be an economic noble-metal catalyst for low-temperature NO removal, although its performance in the presence of $O_2$ is yet to be testified. In summary, the low-temperature oxidation of hydrocarbon in real engine exhaust would require the incorporation of multiple catalytic sites or even multiple catalyst systems, to address the reaction inhibitors.

## 5. Conclusions

In this work we systematically analyzed the inhibition effects of other components on the catalytic oxidation of isooctane on $Pt/CeO_2$ catalysts under the fully simulated exhaust condition (S-GDI). It was found that CO and NO are the two major inhibitors for low-temperature isooctane oxidation. The development of catalysts for hydrocarbon abatement should not only focus on promoting the intrinsic activity for hydrocarbon oxidation, but also on addressing the reaction inhibition by other exhaust components.

The introduction of a low-cost $Cu_1/CeO_2$ single-atom catalyst, which is highly active for low-temperature CO oxidation (<200 °C), even under the fully simulated exhaust condition, successfully eliminates the inhibition effect of CO on the noble metal $Pt/CeO_2$ catalyst for isooctane oxidation. However, the unconverted NO still strongly inhibits the low-temperature oxidation of isooctane. To further eliminate the inhibition effect of NO, so

as to achieve low-temperature hydrocarbon in real engine exhaust, we propose that future work can be focused on exploring catalytic sites for low-temperature $NO_x$ conversion (<200 °C) or the integration of a $NO_x$ removal system into the upstream of the hydrocarbon oxidation catalyst.

**Supplementary Materials:** The following supporting information can be downloaded at: https://www.mdpi.com/article/10.3390/catal13030508/s1, Figure S1: Comparison of isooctane oxidation activity of $Pt/CeO_2+Cu_1/CeO_2$ catalyst mixture under the $O_2+NO+CO$ and S-GDI conditions; Figure S2: Activity of a bi-metallic $Pt/Cu_1/CeO_2$ catalyst.

**Author Contributions:** Conceptualization, Y.W. and F.L.; methodology, Y.W. and F.L.; investigation, F.L. and C.E.G.-V.; writing—original draft preparation, F.L.; writing—review and editing, Y.W., F.L., and C.E.G.-V. All authors have read and agreed to the published version of the manuscript.

**Funding:** This research was funded by US Department of Energy (DOE), Vehicle Technologies Office.

**Acknowledgments:** This research was conducted as part of the Co-Optimization of Fuels & Engines (Co-Optima) project sponsored by the U.S. Department of Energy Office of Energy Efficiency and Renewable Energy, Bioenergy Technologies and Vehicle Technologies Offices. Part of the research described in this paper was performed in the Environmental Molecular Sciences Laboratory (EMSL), a national scientific user facility sponsored by the DOE's Office of Biological and Environmental Research and located at PNNL. PNNL is operated for the US DOE by Battelle.

**Conflicts of Interest:** The authors declare no conflict of interest. The funders had no role in the design of the study; in the collection, analyses, or interpretation of data; in the writing of the manuscript; or in the decision to publish the results.

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
