# Peer review of "A Bifunctional Pt/CeO2-Cu1/CeO2 Catalyst System for Isooctane Oxidation under Fully Simulated Engine-Exhaust Condition: Eliminating the Inhibition by CO"

_catalysts, doi:10.3390/catal13030508_

Round 1
Reviewer 1 Report
Reviewer Comments:
This manuscript focused on reducing the inhibitors of hydrocarbon oxidation catalysts in order to improve the activity of the hydrocarbon abatement catalyst. It is a new idea for catalyst research, and the result is also useful. However, before publication, the following issues need to be revised.
(1) The experimental data of isooctane oxidation under the condition of O2+H2O and O2+CO+NO over the Pt/CeO2-Cu1/CeO2 bifunctional catalyst system and the corresponding description should be added. It will be helpful for the readers to understand the main idea of this manuscript.
(2) Only discussion has limited support to the conclusion. Some experimental data will be more useful.
(3) On page 2 line 50, “On The other hand” should be On the other hand.
(4) Some format error in the references part.
Reviewer 2 Report
In this manuscript, authors systematically analyzed the inhibition effects of CO, NO, and H2O on the isooctane oxidation over Pt/CeO2 catalysts under stoichiometric condition. Authors also achieved the decrease in the oxidation temperature by eliminating the CO inhibition effect using Cu1/CeO2 catalyst. The manuscript is well written. However, there are some issues to be addressed below. Authors are encouraged to resubmit the manuscript after addressing following comments.
Comments:
1. Line 35, in Introduction, it is stated that HC oxidation is inhibited by H2O. Especially, regarding to Figure 2, the authors better address the possibility that H2O as a product inhibits the oxidation of HC.
2. The readers can easily predict that the co-impregnation method is promising. In addition to removing the inhibitor, it is better to describe the desired catalyst structure from the viewpoint of catalyst design for future work.
Minor comments:
1. Line 50, “On The other hand” should be replaced with “On the other hand”.
2. The plot is large and it is not easy to grasp the trend. Therefore, it is better to make the plot smaller and use lines to emphasize the trend.
Round 2
Reviewer 1 Report
The author has revised the manuscript significantly for publication.
I have no problem.